# Trastuzumab–Deruxtecan for the Treatment of Metastatic Breast Cancer Patients: Data from Real World Studies

**DOI:** 10.3390/cancers17213505

**Published:** 2025-10-30

**Authors:** Erica Quaquarini, Federica Luelli, Angioletta Lasagna, Gianpiero Rizzo, Lorenzo Perrone, Simone Figini, Raffaella Achille, Paolo Pedrazzoli

**Affiliations:** 1Department of Oncology, Comprehensive Cancer Center, Fondazione IRCCS Policlinico San Matteo, 27100 Pavia, PV, Italy; federica.luelli01@universitadipavia.it (F.L.); a.lasagna@smatteo.pv.it (A.L.); g.rizzo@smatteo.pv.it (G.R.); l.perrone@smatteo.pv.it (L.P.); simone.figini01@universitadipavia.it (S.F.); r.achille@smatteo.pv.it (R.A.); p.pedrazzoli@smatteo.pv.it (P.P.); 2Department of Internal Medicine and Medical Therapeutics, University of Pavia, 27100 Pavia, PV, Italy

**Keywords:** trastuzumab–deruxtecan, metastatic breast cancer, real-world, HER2-positive, HER2-low

## Abstract

**Simple Summary:**

The purpose of this review is to summarize the background and latest evidence for the use of Trastuzumab–deruxtecan (T-DXd), a new-generation antibody drug conjugate, in advanced HER2-positive/HER2-low breast cancer, with a focus on some of the unanswered questions about the performance of this agent in clinical practice. The available clinical data from both controlled clinical trials and real-world experience with T-DXd in patients with HER2-positive/HER2-low metastatic disease, including subgroup analyses, have been reviewed and discussed. T-DXd significantly improved progression-free survival, clinical benefit rates, and overall survival in randomized clinical trials. The tolerability profile was manageable, with neutropenia being the most common adverse event. Available data from real-life experiences confirm the good performance of T-DXd in unselected, heavily pretreated populations.

**Abstract:**

**Background**: Trastuzumab–deruxtecan (T-DXd), a new-generation antibody drug conjugate, has greatly improved the survival and clinical benefit rates of patients affected by advanced HER2-positive/HER2-low breast cancer according to the results of controlled clinical trials with a manageable safety profile. Data from randomized clinical trials can provide valuable information for the management of patients in everyday clinical practice, including those who would typically be excluded from such trials due to not meeting the inclusion criteria. **Methods**: In this narrative review, we describe and discuss real-world studies in the literature on the use of T-Dxd in HER2-positive and HER2-low MBC patients, providing a critical analysis of the specific settings of clinical interest. **Results**: Using a PubMed search, we identified nine real-world studies on T-DXd that are available in the literature. A total of 7146 patients have been included in these retrospective studies. A total of 5/9 studies also included HER2-low MBC patients. In the majority of cases, patients had high disease burden with lung and liver involvement. We then reviewed and discussed clinical areas of interest, including heavily pretreated patients, poor performance status, HER2-positive versus HER2-low disease, brain metastasis, elderly patients, lung toxicity, safety profile, and dose modifications. **Conclusions**: Our analysis confirms the activity of the drug described in real-world studies and shows a favorable safety profile, with manageable adverse effects.

## 1. Introduction

Breast cancer (BC) is the second most common cancer and the fourth leading cause of cancer-related mortality worldwide [1]. Overexpression and/or amplification of the human epidermal growth factor receptor 2 (HER2) occurs in up to 20% of BC cases and is associated with more aggressive disease and poorer prognosis [2]. Although treatable, metastatic breast cancer (MBC) remains incurable, with a 5-year overall survival (OS) rate of only 31% [3]. Over the past decades, the introduction of new anti-HER2 agents has significantly improved the prognosis of patients with both HER2-positive and HER2-low MBC [4] (Figure 1).

Trastuzumab and pertuzumab, two HER2-targeting monoclonal antibodies, combined with taxane-based chemotherapy, represent the first-line standard of care, achieving a median progression-free survival (mPFS) of 18.7 months in treatment-naïve HER2-positive MBC patients, and a median OS (mOS) of 56.5 months [5]. Upon progression after first-line therapy, antibody-drug conjugates (ADCs) are the preferred therapeutic approach. Trastuzumab–deruxtecan (T-DXd) has recently been approved for use in this setting, following the impressive results of the DESTINY-Breast03 trial [6]. This second-generation ADC demonstrated superiority over trastuzumab emtansine (T-DM1), a first-generation ADC, with a mPFS of 28.8 months (95% CI: 22.4–37.9) versus 6.8 months (range: 5.6–8.2), and a mOS not reached in both arms (95% CI: 40.5 months–not estimable versus 34.0 months–not estimable). Additional clinical trials have further highlighted the strong efficacy of T-DXd in various clinical contexts (Table 1) [6,7,8,9,10].

In comparative studies, T-DXd has shown superior outcomes compared to trastuzumab or lapatinib combined with capecitabine, with a mPFS of 17.8 versus 6.9 months, and a mOS of 39.2 versus 26.5 months [8]. An alternative therapeutic option for second- or third-line treatment, especially in patients with brain metastases, is the combination of tucatinib (a tyrosine kinase inhibitor) with trastuzumab and capecitabine, as demonstrated in the HER2CLIMB trial [11,12]. This combination significantly improved both PFS and OS compared to trastuzumab plus capecitabine alone, not only in the overall population (mPFS of 7.6 versus 4.9 months; mOS of 24.7 versus 19.2 months), but also in the subgroup with brain metastases (mOS 18.1 months). The DESTINY-Breast12 trial further investigated T-DXd in HER2-positive MBC patients with brain metastases [10,13]. In the brain metastases cohort, the 12-month PFS was 61.6% (95% CI: 54.9–67.6), and the 12-month central nervous system (CNS)-PFS was 58.9% (95% CI: 51.9–65.3) demonstrating substantial and durable systemic and intracranial activity of T-DXd and supporting its use in previously treated HER2-positive MBC patients regardless of brain metastasis status. Furthermore, in HER2-low breast cancer, defined as a score of 1+ by immunohistochemistry (IHC) or 2+ with negative in situ hybridization (ISH), promising results have also been reported [9]. The DESTINY-Breast04 trial showed that in the hormone receptor-positive subgroup, T-DXd led to a mPFS of 10.1 months compared to 5.4 months with physician’s choice (hazard ratio [HR] = 0.51; *p* < 0.001), and a mOS of 23.9 versus 17.5 months (HR = 0.64; *p* = 0.003). In the overall population, mPFS was 9.9 versus 5.1 months (HR = 0.50; *p* < 0.001), and OS was 23.4 versus 16.8 months (HR = 0.64; *p* = 0.001). Although the significant improvement in mPFS observed across the DESTINY-Breast clinical trials has been consistent across all patient subgroups—regardless of stratification factors or baseline characteristics—there remains an urgent need to evaluate patients with a broader range of clinical presentations and comorbidities, reflective of those encountered in everyday clinical practice.

This review aims to summarize available real-world clinical data on the use of T-DXd in patients with MBC, including both HER2-positive and HER2-low subtypes, based on published randomized clinical trial data.

## 2. Methods

In this narrative review, we describe and discuss real-world studies in the literature on the use of T-DXd in HER2-positive and HER2-low MBC patients, providing a critical analysis of the specific settings of clinical interest. Our analysis also concentrates on the toxicity profile of the drug in unselected patients. The literature search was primarily conducted using PubMed, given its extensive coverage of biomedical literature. We used a combination of keywords and MeSH terms related to “trastuzumab deruxtecan,” “metastatic breast cancer,” and “real-world studies” to identify relevant articles published up to July 2025. The search strategy included terms such as: “trastuzumab deruxtecan”, “metastatic breast cancer” or “advanced breast cancer”; “real-world evidence” or “observational study” or “registry” or “real-world study” or “real-life studies”. No language restrictions were applied, but only articles published in English were included. To enhance comprehensiveness and reduce publication bias, reference lists of included articles and relevant reviews were screened for additional studies. Inclusion criteria were as follows: studies reporting clinical outcomes, safety, or treatment patterns of T-DXd in adult patients with MBC in real-world settings; studies including adult patients (≥18 years old); publications in English; and studies published within the last 10 years. Exclusion criteria included: preclinical studies, clinical trials, and case reports with fewer than 10 patients, abstracts, review articles without original data, and non-English publications.

## 3. Real-World Studies

We identified nine real-world studies on T-DXd in MBC treatment reported in the literature (Table 2) [14,15,16,17,18,19,20,21,22]. All but one were retrospective multicenter trials; the exception was a single-center study. The enrolled populations and lines of therapy were highly heterogeneous. In total, these studies included 7146 patients. The mean age of patients was 57 years, and all had received at least three prior lines of treatment before starting T-DXd. Five out of nine studies also included patients with HER2-low MBC. Most patients presented with a high disease burden, often involving the lungs and liver. Only one trial specifically focused on patients with brain metastases [21]. The primary objective in all studies was treatment activity, measured by ORR or objective response categories (complete response, partial response, stable disease, and progressive disease), except for the study by Joudain et al., which had OS as its primary objective, and the study by Nakajima and colleagues, which evaluated efficacy in terms of OS, PFS, and 24-month OS [17,18]. All studies assessed treatment safety as a secondary objective.

### 3.1. Heavily Pretreated Patients

Six trials have specifically addressed the evaluation of the activity of T-DXd in heavily pretreated patients. The first study by Fountzilas et al. enrolled 312 patients; 72.4% of them received T-DXd, 19.2% sacituxumab govitecan, and 8.3% both agents [16]. Among patients with HER2-positive MBC, the 12-month PFS rate was 69.6% (95% CI, 61.4% to 79%) and no differences in mPFS were seen between patients receiving T-DXd as first- or second-line therapy compared to those treated in the third line or beyond (17.31 months, 95% CI, 11.21 months to non-estimable versus 15.27 months, 95% CI, 11.79 to 22.04 months; HR 1.07 [95% CI, 0.71 to 1.61]; *p* = 0.752). In the same study, patients with HER2-low MBC had a 12-month PFS rate of 36.5% (95% CI, 28.6% to 46.5%) and mPFS was longer in patients treated as first- or second-line therapy compared to later lines (11.56 months, 95% CI, 9.72–17.71 versus 6.34 months, 95% CI, 4.37–17.31; HR 0.47 [95% CI, 0.29–0.81]; *p* = 0.006). Correspondingly, the 12-month PFS rates were 51.3% (95% CI, 39.5% to 66.8%) for first or second-line treatment and 35.7% (95% CI, 18.5% to 69.1%) for subsequent lines.

In the study by Botticelli et al., 143 patients were included, and T-DXd was administered as first-, second-, third-, or subsequent-line therapy in 3%, 11%, 29%, and 57% of patients, respectively [15]. Patients had received a median of 4 prior metastatic therapy lines (range 1 to 11). A favorable trend was observed in real-world PFS for patients treated with T-DXd as first- or second-line therapy compared with later lines (17 versus 15 months; *p* = 0.098). However, no significant difference in ORR was found between these groups (72% versus 67%; *p* = 0.88). These findings align with results from the DESTINY-Breast01 and DESTINY-Breast02 trials, where T-DXd was administered after varying numbers of prior systemic therapies. The DE-REAL study reported that only a small proportion of patients received T-DXd in the first- or second-line setting, yet their outcomes were comparable to those reported in DESTINY-Breast03, with 12-month PFS of 75.2%, 12-month OS of 94.1%, and ORR of 79%.

Similarly, in the study by Sang et al., 61 patients were included [22]. T-DXd was mainly used in later treatment lines (third or fourth line in 28 cases [45.9%], and beyond the fourth line in 19 cases [31.15%]). Patients who had received no more than one prior chemotherapy line had a mPFS that was not reached (95% CI, 1.58–not estimable), while those with more than one prior chemotherapy line had a shorter mPFS of 10.51 months (95% CI, 1.64–not estimable).

In the study by Bizarro et al., a total of 100 MBC women were included, and all of them had received at least two prior treatments for advanced disease [20]. In particular, T-DXd was administered as third-line therapy in 52% of patients, fourth-line therapy in 15%, and fifth-line therapy or beyond in 23%. Results showed that administration of T-DXd in later lines was associated with a lower ORR (44%) compared to registration trials, such as DESTINY-Breast02, which reported an ORR of 70%. However, the registered CBR was 80%. The mPFS was 13 months (95% CI: 10–16 months), and the 12-month PFS was 54%.

Petit et al. published a real-world study including 459 MBC patients treated with T-DXd under a temporary use authorization program in France [14]. Patients had received a median of 4 prior metastatic treatment lines (range 2–22). Before inclusion in the study, 81.7% had received radiation therapy, and 76.5% had undergone surgery. Among 160 patients with evaluable tumor assessments, the ORR was 56.7%, with 12.1% experiencing disease progression.

In the last study by Lazarotos et al., 38 heavily pretreated patients were included [21]. These patients had received a median of four prior lines of therapy (range 1–12) before starting T-DXd. Among 33 evaluable patients, 21 (63.6%) experienced a response, including 3 complete responses (9.1%) and 18 partial responses (54.5%). Stable disease was observed in five patients (15.2%), and seven patients (21.2%) had progressive disease.

Taken together, these studies consistently demonstrate the robust antitumor activity of T-DXd in heavily pretreated patients, including those with significant comorbidities and a high disease burden—populations often underrepresented in randomized clinical trials. This reinforces the potential of T-DXd as an effective therapeutic option, even in complex real-world scenarios where patients have already undergone multiple lines of therapy. Importantly, the data indicate that the number of prior anti-HER2 treatments does not significantly diminish the sensitivity of HER2-positive MBC to T-DXd. This suggests that resistance mechanisms associated with earlier lines of therapy may not fully compromise the efficacy of this agent. Instead, treatment response appears more closely linked to intrinsic tumor biology, particularly the level of HER2 expression on the tumor cell membranes and the tumor cells’ inherent susceptibility to the cytotoxic payload, DXd.

### 3.2. Poor ECOG PS Patients

The ECOG Performance Status (PS) is a standardized scale broadly used in Clinical Oncology to quantify a patient’s functional status and ability to perform daily activities. It ranges from 0 (fully active) to 5 (dead). It is a critical tool for clinical decisions and prognosis [23].

Of the nine real-world studies considered in this review, only five trials reported the ECOG PS of the patients included, which was favorable (0–2) in the majority of cases.

Nakayama and colleagues published the updated results of the ROSET-BM trial, which included 104 HER2-positive MBC patients with brain metastasis and/or leptomeningeal disease. The study also included patients with poorer ECOG PS, specifically 26% with ECOG PS 0, 51.9% with ECOG PS 1, 11.5% with ECOG PS 2, and 3.8% with ECOG PS 3–4 [18]. In this study, it was reported that background factors—including the presence of leptomeningeal carcinomatosis, components of Graded Prognostic Assessment (GPA) scoring such as age, number of brain metastases, extracranial metastases, ECOG PS, as well as other factors (HER2 status, IHC, estrogen receptor status, steroid use at the time of T-DXd administration, surgery, and treatment line)—were not identified as prognostic factors for OS.

Petit et al. included patients up to ECOG PS 3 (36.4% of ECOG PS 0 patients, 54% of ECOG PS 1 patients, 9.2% of ECOG PS 2, and only 0.4% of ECOG PS 3 patients) [14]. For the only two ECOG PS 3 patients, a reduced dose of T-DXd (4.4 mg/kg) was provided instead of the traditional dose (5.4 mg/kg). No PS-related side effects are reported, and no data on activity were reported.

In the article by Sang et al., both HER2-positive and HER2-low MBC patients were included [22]. Patients were divided according to the initial ECOG PS score (68.85% had ECOG PS 0–1 and 31.15% had ECOG PS 2–4), and 40% of HER2-positive patients had an ECOG PS 2 score and poorer baseline characteristics. Patients with advanced PS received a reduced initial dose, but no data on activity were specifically reported for this subgroup of women.

In Fabi et al.’s article, only ECOG PS 0–2 were included (30.8% ECOG PS 0, 51.3% ECOG PS 1, and 17.9% ECOG PS 2) [19]. However, the activity of T-DXd based on patients’ ECOG PS was not reported. In a study by Bizzaro et al., the median ECOG PS score of the patients included was 1 (range: 0–2), but also in this case, no correlations with activity were reported [20].

Taken as a whole, the available data on patients with poor ECOG PS treated with T-DXd, while limited, are encouraging. Importantly, no unexpected toxicities or worsening of baseline clinical conditions were observed in this vulnerable subgroup. These findings suggest that T-DXd may still be a viable treatment option in patients with impaired functional status—a group typically excluded from registration trials and underrepresented in the pivotal DESTINY-Breast studies. What is particularly noteworthy is the deviation from standard trial protocols observed in real-world clinical practice. Several studies report that patients with poor ECOG PS were started on a reduced initial dose of T-DXd, a strategy not supported by the drug’s prescribing information or evaluated in registration trials. This pragmatic approach appears to be driven by clinical judgment, aiming to balance efficacy with tolerability in patients at greater risk for toxicity due to frailty, comorbidities, or high disease burden. However, there is currently a lack of prospective data on dose modifications for patients with ECOG PS ≥ 2, and no clear guidance on whether starting at a lower dose compromises efficacy or improves safety outcomes.

### 3.3. HER2-Low Versus HER2-Positive

Among the studies analyzed, four of them included patients with HER2-low MBC. The first study by Sang and colleagues compared the activity and safety of T-DXd in a Chinese population of women with both HER2-low and HER2-positive disease [22]. PFS was 10.51 months (95% CI, 3.02–not estimable) in the HER2-low group and 10.18 months (95% CI, 3.88–not estimable) in the HER2-positive group. ORR were 37.93% and 62.50%, disease control rates (DCR) were 79.31% and 87.50%, and median time to response (mTTR) was 1.28 and 1.31 months, respectively. In the HER2-low group, no complete responses were observed, whereas in the HER2-positive group, two patients (6.25%) achieved complete responses. Partial responses were seen in 11 (37.93%) HER2-low and 18 (56.25%) HER2-positive patients, and stable disease in 12 (41.38%) and 8 (25.00%) cases, respectively. Progressive disease occurred in six (20.69%) HER2-low and four (12.50%) HER2-positive patients.

Another study specifically evaluating MBC patients with both HER2-positive and HER2-low disease reported no significant difference in ORR between the two groups, although a trend toward longer PFS was observed in HER2-positive patients; however, this did not reach statistical significance [21].

The largest real-world study published by Jourdain et al. included 5890 patients: 2010 (34.1%) with HER2-positive disease treated in the third line, 1260 (21.4%) treated in the second line, and 2620 (44.5%) with HER2-low disease [17]. Among patients receiving third-line therapy for HER2-positive MBC, the mOS was 30.2 months (95% CI, 28.1–33.5), with a 1-year survival rate of 80.5% (95% CI, 78.7–82.3%). In the second-line HER2-positive group, mOS was not reached, and the 1-year survival rate was 85.6% (95% CI, 83.4–87.9%). Among HER2-low patients, mOS was 16.8 months (95% CI, 14.5–not reached), with a 1-year survival rate of 62.3% (95% CI, 59.7–65.0%). The median time to treatment discontinuation (mTTD) was 10.8 months (95% CI, 10.4–11.5) for third-line HER2-positive patients, 11.7 months (95% CI, 11.0–12.9) for second-line HER2-positive patients, and 5.6 months (95% CI, 5.5–5.9) for HER2-low patients. Among those who discontinued treatment, 58.8% transitioned to trastuzumab monotherapy (594/1011), and 18.9% to oral chemotherapy (191/1011). In the second-line HER2-positive cohort, the median time to first subsequent treatment (mTFST) was longer at 14.6 months (95% CI, 13.8–16.4), with trastuzumab alone as the most common subsequent therapy (45.4%, 189/416). For HER2-low patients, the mTFST was 8.4 months (95% CI, 8.1–8.7), with oral chemotherapy (48.7%, 399/819) and sacituzumab govitecan (38.7%, 317/819) being the most frequently used next-line treatments. Multivariate Cox regression analysis identified increasing age, a higher number of prior treatment lines, presence of digestive or brain metastases, recent diagnosis of BC or metastatic disease, and existing comorbidities as factors associated with an increased risk of death. Notably, the results showed that both HER2-positive and HER2-low groups experienced a significant increase in hospital admissions for respiratory, digestive, and hematological conditions following initiation of T-DXd. However, these hospitalizations may not be solely attributable to T-DXd itself, but also to the underlying metastatic disease, patient comorbidities, and overall cancer-related frailty, all of which contribute to fluctuating clinical status and performance.

The real-world study published by Fountzillas et al. included 122 patients with HER2-low MBC and 128 with HER2-positive disease [16]. T-DXd demonstrated meaningful clinical activity in HER2-low BC patients; however, outcomes were less favorable than in patients with HER2-positive disease. The 12-month PFS rate was 36.5% (interquartile range [IQR], 28.6–46.5) in HER2-low patients and 69.6% (IQR, 61.4–79.0) in HER2-positive patients. As expected, in the HER2-low group, earlier-line treatment with T-DXd was associated with significantly improved PFS compared to later lines (11.56 months, 95% CI 9.72–17.71 versus 6.34 months, 95% CI 4.37–17.31; HR 0.47 [95% CI 0.29–0.81]; *p* = 0.006). For first-/second-line therapy, the 12-month PFS rate was 51.3% (95% CI 39.5–66.8%), whereas they resulted in 35.7% (95% CI 18.5–69.1%) for later lines. Among HER2-positive patients, a 12-month PFS rate of 69.6% was reported (95% CI 61.4–79.0%). No significant differences in mPFS were observed between those treated in the first or second line and those treated in later lines (mPFS 17.31 months, 95% CI 11.21–not estimable, versus 15.27 months, 95% CI 11.79–22.04; HR 1.07 [95% CI 0.71–1.61]; *p* = 0.752).

Taken together, these findings suggest that T-DXd can offer disease control in HER2-low patients, albeit with a more modest response depth, shorter durations of benefit, and a quicker transition to other therapies such as sacituzumab govitecan or oral chemotherapy. These disparities highlight the biological differences between HER2-positive and HER2-low tumors and likely reflect varying levels of HER2 expression that impact drug-target engagement and ADC efficacy. Importantly, across all studies, earlier use of T-DXd (i.e., in first or second line) was associated with improved outcomes in HER2-low patients. This raises a critical consideration for clinical decision-making: while T-DXd remains effective across multiple lines, maximizing its benefit may require earlier integration into the treatment sequence for HER2-low disease. In contrast, in HER2-positive patients, prior anti-HER2 therapies do not appear to markedly diminish T-DXd’s activity, which continues to perform well even in later lines of therapy.

### 3.4. Brain Metastasis

According to the literature, about 30–50% of patients with HER2-positive BC will develop brain metastases. This is due to both an improvement in OS, thanks to new anti-HER2 agents, and to increased detection rates with the latest imaging techniques. However, the central nervous system (CNS) compartment is still challenging, because the blood–brain barrier (BBB) limits penetration of many compounds, including chemotherapy and targeted treatment. This fact reflects the reduced prognosis of patients with brain metastasis. Local therapy, such as whole-brain radiotherapy (WBRT), stereotactic radiotherapy (SRT), stereotactic radiosurgery (SRS), and neurosurgery, has been a mainstay of brain metastasis treatment, but growing evidence in the literature supports the effectiveness and safety of ADCs for MBC with brain metastasis. Preliminary data from the DEBBRAH, a non-randomized single-arm trial, showed promising intra- and extracranial activity of T-DXd in different cohorts of pretreated patients: the first group with stable brain metastases after local treatment; the second group with asymptomatic untreated brain metastases, and the third group with progressive brain disease after local therapy [24]. Moreover, the TUXEDO trial, an open-label, single-arm, phase 2 study, described an intracranial response rate of 73.3% in patients with HER2-positive MBC [25]. Data from the DESTINY-Breast 12 trial showed substantial and durable overall and intracranial activity with T-DXd in patients with HER-2 positive BC and brain metastasis, irrespective of stable/active baseline brain metastasis [10] (Table 1).

Regarding real-life studies, all studies included patients with brain metastases. The work by Fabi and colleagues specifically focused on patients with brain metastasis [19]. The primary endpoint of this last study was the intracranial overall response rate (iORR). Secondary endpoints were intra- and global progression-free survival (iPFS—gPFS); others were the intracranial disease control rate (iDCR), duration of response (iDoR), clinical benefit rate at 6 and 12 months (iCBr), OS, and safety. A total of 39 patients were included, and the results showed an iORR of 59%, an iPFS of 15.6 months, a gPFS of 11.8 months, an iDCR of 94.9%, an iDoR of 11.9 months, and an iCBr at 6 and 12 months of 69.2% and 59%. Overall survival was not reached, with 77.9% of patients alive at 12 months. Outcomes were independent of the treatment line, and the treatment had a manageable safety profile. No differences in mPFS were observed between patients who received or did not receive local treatment for brain metastases (15.8 versus 15.6 months, *p* = 0.45).

Another study specifically evaluated patients with HER-2 positive MBC and brain metastasis and/or leptomeningeal disease (ROSET-BM trial) [18]. In this study, the effectiveness data from a previous publication were updated, and a total of 104 patients were included [26]. The majority had asymptomatic brain metastasis (72 patients, 69.2%) and received a local treatment (99 patients, 95.2%), with only 5 patients (4.8%) untreated for brain metastasis. Local treatment included SRT (64 patients, 61.5%) followed by WBRT (56 patients, 53.8%). Median PFS was 14.6 months, mOS was not reached; 24-month OS rate was 56.0%. Subgroup analysis showed that mPFS was 13.2 months in patients with active brain metastases, 17.5 months in patients with leptomeningeal carcinomatosis, and not reached in patients with stable brain metastases (24-month PFS rates in patients with active brain metastases, leptomeningeal carcinomatosis, and stable brain metastases were 32.7%, 25.1%, and 60.8%, respectively). Median OS was 27.0 months in patients with active brain metastases and not reached in patients with leptomeningeal carcinomatoses or stable brain metastases (24-month OS rates in patients with active brain metastases, leptomeningeal carcinomatosis, and stable brain metastases were 52.0%, 61.6%, and 71.6%, respectively).

In the study by Fountzilas et al., at T-DXd initiation, 59 patients (23.5%) had brain metastases and a mPFS of 13.24 months was registered in this group of patients (either extracranial or intracranial sites) [16]. Central nervous system relapse occurred in 10 patients (18.5%) during T-DXd treatment, among whom 7 (16.7%) had HER2-positive and 3 (25%) had HER2-low disease.

In the study by Sang et al., 7 (24.14%) HER2-low patients and 10 (31.25%) HER2-positive cases had brain metastasis [22]. The ORR was 28.6%, with more than 50% of these patients having HER2-low expression. Among patients diagnosed with HER2-positive MBC and brain metastases, those who have undergone no more than three lines of previous treatment had longer PFS (up to 10.55 months, 95% CI, 6.37–NE). By contrast, patients who had received more than three prior lines of treatment reported a significantly shorter mPFS (up to 3.88 months; 95% CI, 1.81–10.18). The same trend has also been described among HER2-low MBC patients.

In the trial by Lanzarotos et al., 15 patients with CNS metastases were included and, at the time of T-DXd initiation, all of them were treated with local modalities (radiation and/or surgery) [21]. Of these 15 patients, 10 patients received brain radiotherapy (66.7%) and 5 patients underwent both surgery and brain radiotherapy (33.3%). A total of 6/15 had progression of brain lesions after previous local treatment and prior to initiating T-DXd (40%), two patients had a complete response (13.3%), 7 a partial response (46.7%), one patient a stable disease (66.7%), and another one a progressive brain and leptomeningeal disease (6.7%). No significant difference in terms of ORR was seen in patients with HER2-positive compared to the HER2-low group. Two patients had no brain metastases at the start of T-DXd but developed leptomeningeal metastases during treatment (5.3%). In patients with brain lesions, the mPFS was not reached; mOS was 420 days, and no significant differences in PFS or OS were observed between patients with HER2-positive versus HER2-low MBC. Results also showed no significant differences in PFS or OS between patients with and without brain metastases.

In another study, a total of 1079 patients with HER2-positive BC and 419 patients with HER2-low BC and brain metastases were included [17]. Multivariate Cox Regression showed that the presence of brain metastasis was an independent predictor for either survival or cause-specific hospitalization, as well as the presence of comorbidities.

In the large trial of Petit and colleagues, in 57 patients with available brain tumor assessment, complete or partial intracranial response was reported for 35.7% patients, and 5.4% had progression [14].

In the study by Botticelli et al., 36 (25%) of the patients had brain disease, but the outcomes were not specifically reported for this subgroup population [15].

Taken as a whole, these data indicate that, in real-world clinical practice, locoregional treatments—such as surgery or radiotherapy—remain the standard initial approach for managing brain metastases from HER2-positive BC before starting treatment with T-DXd. This strategy reflects the current clinical consensus aimed at controlling intracranial disease and alleviating neurological symptoms prior to systemic therapy. However, emerging evidence suggests that in selected patients without active brain metastases, locoregional interventions may be safely deferred, allowing T-DXd to be initiated earlier. This approach offers the potential benefit of minimizing treatment-related morbidity while maintaining effective systemic control. Such a strategy may be particularly relevant for patients with stable or treated brain lesions who can be closely monitored for intracranial progression. Importantly, clinical practice data consistently demonstrate an excellent therapeutic response to T-DXd, even in the context of CNS involvement. Patients treated with T-DXd show significant disease control and symptom improvement, underscoring the drug’s efficacy beyond extracranial disease sites. Moreover, no new or unexpected adverse events have been reported in these real-world settings, reinforcing the favorable safety profile of T-DXd observed in clinical trials. These findings support the evolving role of T-DXd as a vital systemic therapy option for HER2-positive BC patients with brain metastases, highlighting the need for individualized treatment planning. Future research should aim to clarify the optimal timing and sequencing of locoregional versus systemic therapies to maximize patient outcomes, as well as to better understand the long-term safety and efficacy of T-DXd in this complex patient population.

### 3.5. Elderly

The elderly population definitely requires special attention during cancer treatment and also the initiation of new treatments due to the intrinsic frailty associated with these patients. The TREX-Old retrospective multicenter European registry specifically evaluated the toxicity of T-DXd in elderly patients (≥70 years) [27]. However, the study is still ongoing, and only preliminary results have been published. However, from the analysis of the studies we have considered, little data has been available. In the study by Fountzilas et al., there was no significant difference in grade 3/4 toxicity between patients aged <70 years and those aged ≥70 years, nor in dose reduction rates or discontinuation rates [16]. In their study, Journain and colleagues noted that patients in their cohort receiving T-DXd were older than those included in randomized clinical trials, with a median age of 60 years for HER2-positive patients and 61 years for HER2-low patients [17]. Furthermore, 13.8% of patients in that cohort had a cardiovascular comorbid condition, either active or under surveillance, whereas the randomized clinical trials excluded patients with active cardiovascular disease. Diabetes and respiratory conditions represented 9.6% and 7.6%, respectively, of the population using T-DXd.

These data suggest that T-DXd can be safely administered to older patients without a disproportionate increase in severe adverse events. However, the retrospective nature of these results and the potential selection bias—where fitter elderly patients are more likely to receive therapy—must be considered. Prospective studies with comprehensive geriatric assessments are necessary to confirm these findings and better guide treatment decisions.

### 3.6. Interstitial Lung Disease (ILD)

Interstitial lung disease (ILD) refers to a group of lung conditions that cause inflammation and scarring (fibrosis) of the lung tissue, which can impair oxygen exchange. ILD is a known and potentially serious adverse effect of T-DXd. In randomized phase III clinical trials, ILD incidence is estimated to be approximately 12.0%, with most being low-grade (13.6%, 10.4%, and 15.2% in the DESTINY-Breast01, -Breast02, and -Breast03 studies, respectively) [6,7,8].

In the real-life study by Fountzilas E et al., ILD was observed in 17 patients (6.7%), and 1.2% cases experienced grade 3/4 toxicity [16]. All patients had received a median of four prior lines of therapy (excluding T-DXd) before the ILD diagnosis. In the study, follow-up with computed tomography (CT) scans was conducted at a median interval of 13 weeks. The median time from treatment initiation to ILD diagnosis was 4.58 months (IQR 2.48–8 months). Hospitalization for pneumonitis was required in 9 patients (56.2%), and ILD was resolved in 15/17 patients (88.2%) using steroid treatment. Dose reduction was necessary for three patients (1.2%), while T-DXd was definitively discontinued in 8 patients. In two patients with grade 2 ILD, the drug was re-administered after symptom improvement, with no subsequent ILD recurrence. There was one death possibly related to ILD in a patient receiving T-DXd.

In other studies, the incidence of ILD was reported as 2% in the study by Botticelli et al., 3.7% in the study by Petit et al., and 10% in the study by Sang and colleagues, with no ILD-related deaths reported [14,15,22].

In the study of Nakayama et al. ILD was reported in 23.1% of cases and was the most common reason for discontinuation of T-DXd [18]. The median onset of ILD was 5.3 months (95% CI 4.0, 8.8). The incidence of grade 1 ILD (14 cases, 13.5%) was higher than ILD of other grades (grade 2, 3 cases, 2.9%; grade 3, 5 patients, 4.8%; grade 4, 2 cases, 1.9%). No cases of ILD-related death were described.

In the study by Bizarro et al., any-grade ILD was reported in 12% of cases (with 3% ≥ grade 3) [20]. Regarding ILD of any grade, seven (58%) needed oral steroid therapy, and five (42%) intravenous steroid therapy. One patient worsened even with invasive mechanical ventilation and died.

In the study by Lazarotos and colleagues, four patients experienced pneumonitis ILD, and three patients (7.9%) discontinued T-DXd treatment due to grade 2 pneumonitis/ILD [21].

In the study by Jourdain et al., ILD was reported in 36 patients (1.1%) with HER2-positive BC and in 23 patients (0.9%) with HER2-low BC. Although the severity grade was not specified, hospitalization was required in 1.1% of cases [17].

In conclusion, ILD is a recognized and significant risk included in the risk management plan for T-DXd. Real-world data have shown variable ILD incidence, though not higher than that reported in randomized clinical trials. Increasing evidence has also been published regarding clinical management strategies. Close monitoring of patients is essential to prevent high-grade ILD, particularly through the use of CT scans and prompt investigation of early respiratory symptoms. According to current adverse event management guidelines, permanent discontinuation of T-DXd is recommended for grade 2 or higher ILD [28]. Rechallenge may be considered in patients who develop grade 1 or 2 ILD, with possible dose adjustments to balance the benefit–risk ratio.

### 3.7. Safety and Dose Modifications

Adverse events were reported in all studies analyzed, except in the work by Nakayama et al., in which only ILD incidence is reported. The most common adverse events were asthenia (about 9%), nausea (about 5%), and neutropenia (about 8%), but G3-G4 events were rare (Table 2). The incidence of alopecia varies among the studies, with a minimum incidence of 2.5% in the study by Petit et al. and a maximum incidence of 59% in the study by Fabi et al. [14,19]. These differences may depend on the type of patients and, in particular, on the types of previous treatments.

In the study by Botticelli et al., the reported rate of toxicities and severe toxicities was lower than in published trials [15]. In particular, despite the enrollment of patients with an older median age, the authors reported lower bone marrow toxicity (i.e., anemia, decreased neutrophil count) when compared to the observed rates of bone marrow toxicities in published clinical trials and a lower incidence of nausea/vomiting. This finding may depend on several factors: the first, an under-reporting of the incidence and grading of adverse events in retrospective studies; second, the ability to prevent or manage T-DXd-related toxicities in clinical practice thanks to the results of randomized clinical trials; last, in real-life studies, dose reductions are more frequent due to less stringent protocols that may result in lowered toxicities. Furthermore, subjective toxicity assessment in real-life studies may be affected by underreporting in medical records, as already shown in the literature [29]. However, in this trial, the ORR was comparable in patients with and without any-grade toxicity (67% versus 68%, respectively, *p* = 0.74) and with low- and high-grade toxicities (68% versus 66%, respectively, *p* = 0.63).

In the study by Sang et al., according to ECOG PS scores, weight, and economic status, the majority of patients (54 patients, 88.52%) received a reduced initial dose of T-DXd, whereas 5 patients (8.20%) had dose adjustments during treatment due to gastrointestinal and bone marrow side effects [22]. However, adverse reactions occurred in 59 patients, and the most frequent ones were nausea in 48 patients (78.69%), anorexia in 45 (73.77%), leukopenia in 21 (34.43%), anemia in 18 (29.51%), alopecia in 14 (22.95%), vomiting in 12 (19.67%), diarrhea in 9 (14.75%), thrombopenia in 9 (14.75%), and constipation in 6 (9.84%).

In the trial by Bizarro and colleagues, adverse events of any grade were present in 83 women [20]. The most frequent events were nausea in 49 patients (49%), neutropenia in 37 (37%), and alopecia in 34 (34%). Grade 3 or higher events were present in 16 patients; 10 (10%) had neutropenia, 3 (3%) had fatigue, and 3 (3%) had ILD. Reduced ejection fraction of any grade was seen in 5% of patients. Dose reduction or treatment discontinuation due to toxicity events was required in 46 patients (46%).

In the study by Petit et al., a total of 97/459 treated patients experienced adverse events, and 41 were considered serious [14]. Thirteen fatal cases were reported (three related to treatment; nine unrelated; one unknown). The causes of the three treatment-related deaths were: altered general condition with neurological decompensation and weight loss; lung and cardiac disorders (not ILD); and no information was reported on the cause of death of the third case. The most common reported adverse events were: nausea (grade 1–2, 17.7%, grade ≥ 3, 5.5%), neutropenia (grade 1–2, 4.5%), asthenia (grade 1–2, 10.1%), and decreased appetite (grade 1–2, 2.5%, grade ≥ 3, 0.5%). For the 459 patients included, the planned dosage was the standard dose for 452 patients (98.5%) and a level—1 reduced dose (4.4 mg/kg) for 6 cases for the following reasons: abnormal liver function tests (n = 2), low weight (n = 1), occurrence of adverse event during the previous treatment and comorbidities (n = 1) and ECOG PS 2 (n = 1). The dosage was planned at 3.2 mg/kg for one patient due to poor tolerance of different previous chemotherapies. For 39 patients, the treatment was discontinued due to: cancer progression (n = 14), death (n = 13), patient’s wish (n = 4), adverse events (n = 3), disease progression/adverse reaction/patient’s wish (n = 1), and other (n = 2). Seventeen end-of-treatment values were reported (12 missing data), and 21 changes in dose were reported (11 missing data).

The study by Jourdain and colleagues showed an increased hospitalization for HER2-positive and HER2-low groups for respiratory, digestive, and hematological disorders after T-DXd initiation compared to the prior period [17]. However, the reasons for admission to hospital may not be entirely related to T-DXd administration but also to the underlying metastatic disease and associated comorbidities. Due to the nature of the database used in the study, the authors lacked access to data on treatment delivery, toxicity, treatment discontinuation, and disease progression. Also, no modification of the dose was reported due to toxicity.

In the study by Lazarotos, only 3/38 patients stopped T-DXd due to grade 2 pneumonitis, 14 had delayed drug infusion, and 12 had a dose reduction due to adverse events. Posology of the reduction is not specified in the paper [21].

In conclusion, T-DXd has shown a manageable safety profile in the studies analyzed, with most adverse events being consistent with those typically observed with antibody–drug conjugates and in randomized clinical trials. The most commonly reported non-ILD toxicities include gastrointestinal symptoms (such as nausea, vomiting, and decreased appetite), hematologic abnormalities (notably neutropenia and anemia), and fatigue. These events are generally low to moderate in severity and can often be managed effectively with supportive care or dose modifications. However, important limitations emerge. The wide variability in adverse event reporting—such as alopecia rates ranging from 2.5% to 59%—suggests inconsistencies in data collection and highlights the challenges of retrospective study designs. Underreporting, non-standardized toxicity grading, and incomplete documentation are recurrent issues, likely leading to an underestimation of true toxicity rates. Moreover, real-world practices, including frequent dose reductions or initial dose modifications (e.g., due to frailty or ECOG PS), are not reflected in registration trials. These adaptations may contribute to reduced toxicity but complicate efficacy comparisons. The increased hospitalizations observed in some cohorts further stress the need to disentangle treatment-related effects from the natural course of advanced disease and comorbidities.

## 4. Discussion

The introduction of ADCs, and particularly T-DXd, is widely regarded as one of the most significant advances in recent years for the management of HER2-positive BC. ADCs uniquely combine antigen specificity with potent cytotoxic activity, resulting in powerful and highly selective pharmacodynamic effects [30]. T-DXd is a HER2-targeted ADC characterized by a high drug-to-antibody ratio (DAR), which enables the delivery of a substantial concentration of its cytotoxic payload—DXd, a potent topoisomerase I inhibitor—directly to tumor cells. The drug is released through a stable linker that is specifically cleaved by lysosomal enzymes once the ADC is internalized into the target cell. Moreover, the membrane-permeable nature of DXd allows for a bystander effect, enabling it to act on adjacent tumor cells regardless of HER2 expression. This mechanism sets T-DXd apart from other approved HER2-directed ADCs and explains its demonstrated activity in HER2-low and heterogeneous tumors in addition to HER2-positive tumors.

The significant improvement in mPFS observed across the DESTINY-Breast clinical trials has been consistent across all patient subgroups, irrespective of stratification factors or baseline characteristics. Several ongoing randomized clinical trials are currently evaluating T-DXd in various metastatic settings, with preliminary results already available (Table 3) [31,32,33,34].

While clinical guidelines are predominantly based on randomized trial data, real-world clinical practice often involves patients with a broader range of clinical presentations and comorbidities. In this context, the growing body of real-world evidence offers valuable insights into the performance of T-DXd in routine clinical settings. This is the first review in the literature to examine real-world data on the use of T-DXd in patients with HER2-positive and HER2-low BC. Our analysis confirms that real-world outcomes are largely consistent with those reported in pivotal phase III trials. As expected, treatment activity and efficacy correlate with the line of therapy in which T-DXd is administered. For example, Botticelli et al. reported the highest ORR (68%) in the HER2-positive group, while the longest PFS was documented by Bizzaro et al. (13 months) and Fabi et al. (iPFS 15.6 months) [15,19,20]. The most favorable OS was reported by Joudain et al., with a median OS of 30.2 months in HER2-positive patients and 16.8 months in HER2-low patients [17]. Moreover, treatment efficacy seems to correlate also with HER2 expression since patients with HER2-low disease demonstrated modest response depth, shorter durations of benefit, and quicker transition to other therapies than patients with HER2-positive MBC.

In terms of safety, the adverse events reported in real-world studies mirrored those seen in clinical trials, with nausea, vomiting, fatigue, and neutropenia being the most frequently observed toxicities. ILD showed considerable variability in incidence across studies, ranging from 1.2% in Fountzilas et al. to 23.1% in the study by Nakayama et al., confirming that it is important to closely monitor patients through the use of CT scans and prompt investigation of early respiratory symptoms [16,18]. No specific subgroup—based on age, prior treatments, disease location, or extent—was identified as being at differential risk for outcomes or adverse events. Also, in patients with brain metastases, T-DXd demonstrated significant activity and a safe profile, suggesting that, in selected patients without active brain metastases, locoregional interventions may be safely deferred, allowing T-DXd to be initiated earlier. Few data are available for elderly patients but results from the real-life trials analyzed showed that T-DXd can be administered safely in older patients without an increase in adverse events.

Several areas require further research. First, optimizing the use of T-DXd and managing its toxicities is essential. This includes the development of strategies for rechallenging patients who have experienced asymptomatic or fully resolved grade 2 ILD. Additionally, efforts should focus on improving prophylaxis and treatment of T-DXd–induced nausea and vomiting, with particular attention to delayed-onset symptoms. To optimize treatment and personalize care, a more systematic approach is likely needed to identify reliable biomarkers of response and resistance, as well as to define the most effective sequencing of therapies to maximize patient benefit.

Despite their utility, real-world studies have some limitations. A common issue is missing or incomplete data, which can lead to biases and imbalances in analysis, and the retrospective nature of the studies increases the risk of information bias and confounding factors. Furthermore, relatively short follow-up periods may underestimate the true incidence and spectrum of adverse events, especially those that occur late. Another limitation is the lack of randomization and the absence of centralized tumor assessments and standardized response criteria can affect the consistency and reliability of efficacy evaluations, as assessments may vary between institutions or investigators. Moreover, heterogeneity across patient populations, treatment regimens, and healthcare settings may introduce variability that complicates the interpretation of findings and their generalizability. Highlighting the limitations of real-world data is necessary to contextualize the findings, clarify the scope of real-world evidence, and underscore the need for cautious interpretation when comparing with clinical trial data. Nevertheless, the consistency between real-world findings and those of registration trials reinforces confidence in the efficacy and safety of T-DXd in the context of MBC. Moreover, real-world studies reflect treatment outcomes in more diverse and complex patient populations—often older, with comorbidities or more advanced disease—who are typically underrepresented in clinical trials and who often experience poorer outcomes.

Currently, no real-world data are available regarding patient-reported outcomes or the treatment of oligometastatic disease, both of which represent important gaps in the literature. Another key area for advancement is the identification of biomarkers that can predict response to T-DXd therapy. Among the promising tools under investigation is HER2-targeted positron emission tomography (PET), which could help visualize HER2-low metastatic lesions and guide personalized treatment strategies [35].

Considering the positive findings from clinical research, it is likely that T-DXd will be used earlier in the therapeutic sequence for BC. It is currently being studied as a first-line treatment in the metastatic setting (DESTINY-Breast09) and as an adjuvant or neoadjuvant treatment for early-stage disease (DESTINY-Breast05 and DESTINY-Breast11). Given its high efficacy, extended treatment durations are expected, underscoring the need for more rigorous symptom management to preserve patient quality of life.

## 5. Conclusions

T-DXd has been shown to improve treatment outcomes in women with metastatic HER2-low or HER2-positive BC. Given that T-DXd is a relatively recent addition to the therapeutic landscape, valuable insights can be gained by examining its real-world use, monitoring practices, and management strategies across healthcare institutions with experience in administering the drug. Our analysis confirms a favorable safety profile, with adverse effects that are generally manageable and rarely lead to treatment discontinuation. Nonetheless, critical challenges remain in toxicity management, biomarker development, and integration into earlier treatment settings. Addressing these gaps will be paramount to maximizing patient benefit and individualizing therapy in this evolving landscape.

## Figures and Tables

**Figure 1 cancers-17-03505-f001:**
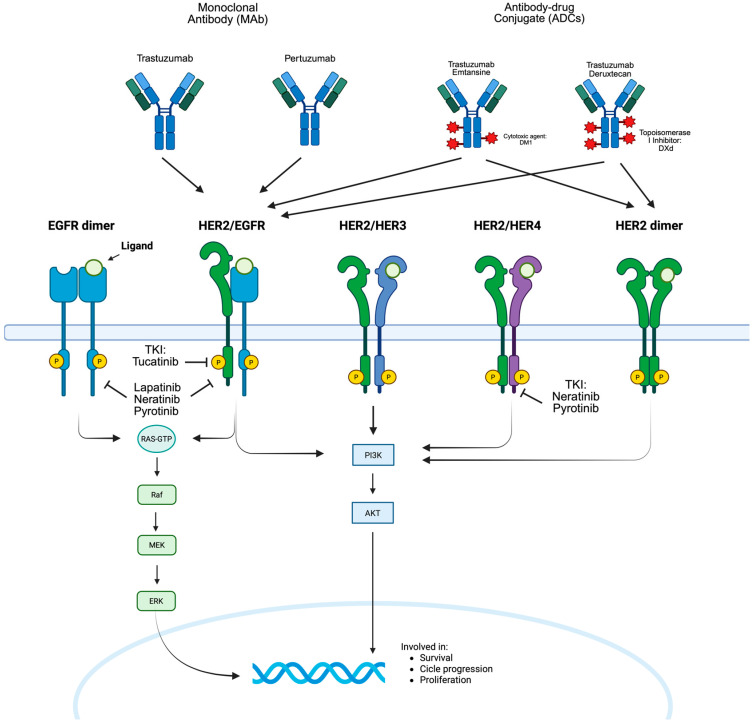
Figure shows how extracellular growth factors and intracellular signals can stimulate cell replication. In particular, the various receptors of the epidermal growth factor receptor family are also described, along with the approved drugs for the treatment of HER2-positive breast cancers and their respective mechanisms of action. Figure edited from Biorender, with copyright permission (Created in BioRender. Luelli, F. (2025) https://BioRender.com/w47f0s6 (accessed on 15 June 2025)).

**Table 1 cancers-17-03505-t001:** Published randomized phase III clinical trial with T-DXd in the metastatic breast cancer setting.

	Destiny Breast 01 [7]	Destiny Breast 02 [8]	Destiny Breast 03 [6]	Destiny Breast 04 [9]	Destiny Breast 12 [10]
Study design	Phase II Open label Single group	Phase III Open label Randomized 2:1	Phase III Open label Randomized 1:1	Phase III Open label Randomized 2:1	Phase IIIb/IV Open label Single arm, two cohorts
Treatment line	≥3	≥3	2°	≥2 (1 or 2 before)	≥2
Study arms	T-DXd (after T-DM1)	T-DXd versus TPC	T-DXd versus T-DM1	T-DXd versus TPC	T-DXd
Primary endpoint	ORR	PFS	PFS	PFS	PFS, ORR
Secondary endpoints	PFS, OS, RR, Response duration	OS, ORR, DoR, safety	OS, ORR, safety	OS (in HR positive and study population), ORR, safety	(CNS) PFS, ORR, time to second PD, (CNS) ORR, new symptomatic BM, TTP, DoR, OS, safety
Visceral metastasis	91.8%	78%	70%	\\	\\
Bone-only disease	\\	\\	\\	\\	With BM: 36.9%, No BM: 35.3%
Liver metastasis	54.9%	\\	\\	70%	With BM: 22.1%, No BM: 27.4%
Brain metastasis	13%	18%	15.6%	6.4%	52.2%
Age < 65 years	\\	79.8%	79.8%	76.5%	\\
Age ≥ 65 years	23.9%	20.2%	20.2%	23.5%	\\
Median Age	\\	54.8 years	54.4 years	57 years	With BM: 52 years, No BM: 54
HER2 3+	83.7%	80%	88.9%	\\	With BM: 71.1%, No BM: 58.5%
HER2 2+	15.2%	19%	10.4%	\\	With BM: 0.8%, No BM: 2.1%
HER2 2+ (low)	\\	\\	\\	42%	\\
HER2 1+ (low)	\\	\\	\\	58%	\\
HER2 1+,2+, ISH positive	15.2%	\\	\\	\\	\\
HER2 status missing	1.1%	\\	<1%	\\	\\
Median number of previous lines	6	2	2	3	With BM: >1 in the metastatic setting.
HR+	52.7%	59%	50%	88.7%	With BM: 62.7%, No BM: 62.2%
HR−	45.1%	41%	50%	11.3%	\\
HR Unknown	2.2%	\\	\\	\\	\\

Abbreviations: BM, brain metastasis; CNS, central nervous system; DoR, duration of response; HR, hormone recepetor; ISH, in situ hybridization; ORR, overall response rate; OS, overall survival; PD, progressive disease; PFS, progression free survival; RR, response rate; T-DXd, trastuzumab–deruxtecan; T-DM1, trastuzumab emtansine; TPC, physician choice of treatment; TTP, time to progression; \\, no data available.

**Table 2 cancers-17-03505-t002:** Real-world studies with T-DXd in the metastatic breast cancer setting.

Author (Ref.)	Type	Pts N/Mean Age	Number of Previous Lines	Treatment Line	HER2-Low	Metastatic Sites	Outcome	Toxicity (Any Grade)	Toxicity (G3–G4)
Petit et al., 2023 [14]	Retrospective, multicenter	459/58	2: 21.1% 3: 19.6% 4: 14.2% 5: 14.6% ≥6: 30.5% Median (range) 4	NA	IHC 2+/ISH−: 1, 0.2% IHC 2+/ISH NA: 3, 0.7% IHC 1+/ISH NA: 1, 0.2% Tot: 1.1%	Bone: 57.3% Lymph nodes: 51.6% Lung: 36.2% Liver: 33.1% Brain: 28.1% Other: 15.3% Cutaneous/subcutaneous: 13.9%	iORR: 35.7% iPD: 5.4% ORR: 56.7% PD: 12.1%	Nausea: 20.2% Fatigue: 12.1% Vomiting: 6.5% Neutropenia: 4.5% Anemia: 4% Diarrhea: 3.5% Alopecia: 2.5% Constipation: 0.5%	ILD: 4.5% Infection and infestation: 4.5% Nausea: 2.5%
Botticelli et al., 2024 [15]	Retrospective, multicenter	143/66	0: 4, 3% 1: 16, 11% 2: 42, 29% ≥3: 81, 57%	NA	NA	Visceral: 59% Brain: 2.5%	CR: 6% DCR: 93% ORR: 68% PD: 7% PR: 62% SD: 25%	Nausea: 32% Fatigue: 20% Neutropenia: 20% Decrease platelet count: 8% Anemia: 6% Alopecia: 6% Increased liver enzymes: 5% ILD: 2% Diarrhea: 1%	Neutropenia: 10% ILD: 2% Anemia: 0.6% Nausea: 1.3% Diarrhea: 0.5% Fatigue: 0% Decrease platelet count: 0% Alopecia: 0% Increased liver enzymes: 0%
Fountzilas et al., 2024 [16]	Retrospective, multicenter	312/51	1: 14, 4.5% 2: 70, 22.5% ≥3: 227, 73%	≥3: 227, 73%	49.2%	Liver: 47.2% Lungs: 36.1% Bone: 36.1% Nodes: 27.8% Brain: 23.5%	CBR: 55.6% ORR 29.2% PFS 5.7 mo	Fatigue: 28.2% Nausea: 25.8% Vomiting: 13.9% Leucopenia: 9.9% Anemia: 8.7% Diarrhea: 8.3% Interstitial lung disease: 6.7% Neutropenia: 5.9% Infection: 1.6% Dizziness: 1.6% Stomatitis: 1.2%	Fatigue: 1.6% Leucopenia: 1.6% Nausea: 1.6% Neutropenia: 1.6% Vomiting: 1.6% ILD: 1.2% Anemia: 0.8% Diarrhea: 0.8%
Jourdain et al., 2024 [17]	Retrospective, multicenter	5890/59	0: 95, 1.6% 1: 1109, 18.8% 2: 1796, 30.5% 3: 1356, 23% 4+: 1534, 26%	NA	44.5%	Digestive metastases: 43.2% Brain: 25.4%	**HER2-positive:** mOS: 30.2 mo **HER2-low:** mOS:16.8 mo	**HER2-positive:** Hematological disease: 7.2% Headache, pain, and fatigue: 6.9% Brain-related disorders: 3.1% Ascites: 1.7% Nausea/vomiting: 1.4% **HER2-low:** Hematological disease: 8.1% Headache, pain, and fatigue: 7.1% Ascites: 3.3% Brain-related disorders: 1.4% Nausea/vomiting: 1.2%	NA
Nakayama et al., 2024 [18]	Retrospective, multicentric	104/NA	0–2: 24% ≥3.76%	NA	NA	Brain: 100% Visceral metastases: 76%	**OS:** All NR Active BM: 27 mo All 14.6 mo Active BM: 13.2 mo 24 mOS; 56% TTF 9.3 mo	NA	ILD: 23.1%
Fabi et al., 2025 [19]	Retrospective, multicenter	39/55 years (35–72)	0: 5.1% 1: 38.5% 2: 25.6% 3: 20.5% >4: 10.3%	0: 5.1% 1: 30.8% 2: 20.5% 3: 20.5% >4: 23.2%	0	Brain: 100%	iCRr: 69.2% (at 6 mo) iCRr: 59% (at 12 mo) iDCR: 94.9% iDoR: 11.9 mo iORR: 59% iPFS: 15.6 mo OS: NR mPFS: 11.8 mo	Alopecia: 59% Fatigue: 53.8% Nausea: 46.1% Neutropenia: 35.9% Constipation: 30.7% Diarrhea: 28.2% Anemia: 23.1% Vomiting: 10.2 Drop of LVEF: 2.5%	Fatigue: 18% Neutropenia: 15.3% Diarrhea: 10.2% Nausea: 7.7% Anemia: 5.1% Mucositis: 5.1% Thrombocytopenia: 2.5% Increase transaminase: 2.5% Pneumonitis: 2.5% Vomiting: 2.5%
Bizarro et al., 2025 [20]	Retrospective, multicentric	100/53.9	1: 10% 2: 52% 3: 15% 4: 6% 5: 6% 6: 6% 7: 5%	≥3	NA	Visceral: 72% Nodes: 69% Bone: 61% Liver: 56% Lung: 54% Brain: 21% Skin: 21%	CBR: 80% CR: 8% mOS: NR ORR: 44% mPFS: 13 mo PD: 20% PR: 36% SD: 36%	Nausea: 49% Neutropenia: 37% Alopecia: 34%	Not specified: 16%
Lazarotos et al., 2025 [21]	Retrospective, single-center	38/57	<4: 42.1% ≥4 57.9%	NA	60.5%	Bone: 68.4 Lung: 50% Liver: 47.4% Brain: 39.5%	CR: 9% OS: 14 mo PFS: 10 mo PD: 18.4% PR: 63% SD: 13.2% **HER2-positive:** CR: 20% PD: 13.3% PR: 33.3% SD: 20% **HER2-low:** CR: 0% PD: 21.7% PR: 56.5% SD: 8.7%	Nausea/vomiting: 63% Fatigue: 55.6% Diarrhea: 25.9% Alopecia: 18.5% Neuropathy: 18.5% Neutropenia: 14.8% Pneumonitis: 14.8% Anemia: 7.4% Thrombocytopenia: 7.4%	Non-specified: Grade 3: 15.8% Grade 4: 0%
Sang et al., 2025 [22]	Retrospective, multicenter	61/55	0–1: 22.95% 2–3: 45.9% ≥4: 31.15%	NA	47.5%	Visceral: 85.25% Liver: 55.74% Lung: 49.18% Brain: 27.87%	**HER2-low** CR: 0% DCR: 79.31% ORR: 37.93% PFS: 10.51 mo PD: 20.69% PR: 37.93% SD: 41.38% TTR: 1.28 mo **HER2-positive** CR: 6.25% DCR: 87.5% ORR: 62.50% PFS: 10.18 mo PD: 12.5% PR: 56.25% SD: 25% TTR: 1.31 mo	Nausea: 78.69% Anorexia: 73.77% Leukopenia: 34.43% Anemia: 29.51% Alopecia: 22.95% Vomiting: 19.67% Diarrhea: 14.75% Thrombopenia: 14.75% Constipation: 9.84% Pneumonia: 1.64%	Leukopenia: 8.20% Anemia: 6.56% Thrombocytopenia: 1.64%

Abbreviations: AI aromatase inhibitor; CBR clinical benefit rate; CR, complete response; DCR, disease control rate; ICBr, clinical benefit rate at 6 and 12 months; iDCR, intracranial disease control rate; iDoR, intracranial duration of response; ILD, interstitial lung disease; iORR, intracranial overall response rate; iPD, intracranial progression disease; iPFS, intracranial progression free survival; mo: months; mOS, median overall survival; mPFS, median progression free survival; NA, not available; NR, not reached; ORR, overall response rate; PD, progressive disease; PR, partial response; SD, stable disease; TTF, time-to-treatment failure.

**Table 3 cancers-17-03505-t003:** Ongoing randomized phase III clinical trial with T-DXd in the metastatic breast cancer setting.

	Destiny Breast 06 (End in 2026) [31]	Destiny Breast 07 (End in 2030) [32]	Destiny Breast 08 (End in 12/2025) [33]	Destiny Breast 09 (End in 2029) [34]
Study design	Phase III Open label Randomized 1:1	Phase Ib/II	Phase I Non randomized	Phase III Open label Interventional Randomized
Treatment line	≥2° (1 or 2° before)	2° (prior anti HER2 regimen)	≥2°	1°
Study arms	T-DXd versus investigator choice chemo in HER2-low and ultralow, HR positive in PD after hormone therapy	T-DXd combination with other anticancer agents in HER2-positive MBC	T-DXd combination in HER2-low advanced or MBC	T-DXd with pertuzumab-matching placebo versus T-DXd with pertuzumab versus standard of care (docetaxel or paclitaxel, Trastuzumab and pertuzumab)
Primary endpoint	PFS	AEs, safety, tolerability	AEs, SAEs	PFS
Secondary endpoints	PFS in intent-to-treat, OS, ORR, DoR, safety	OS, ORR, PFS	ORR, PFS, DoR, OS, immunogenicity of T-DXd	OS, ORR, DoR, PFS2, pain progression, symptoms, tolerability, T-DXd and pertuzumab serum concentration, T-DXd immunogenicity, safety

Abbreviations: AEs, adverse events; DoR, duration of treatment; HR, hormone receptor; MBC, metastatic breast cancer; ORR, overall response rate; OS, overall survival; PFS, progression free survival; PFS2, progression free survival to 2° line of treatment; SAEs, serious adverse events; T-DXd, trastuzumab–deruxtecan.

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
