# Peer review of "Trastuzumab–Deruxtecan for the Treatment of Metastatic Breast Cancer Patients: Data from Real World Studies"

_cancers, 2025, doi:10.3390/cancers17213505_

Round 1

Reviewer 1 Report

Comments and Suggestions for Authors

This manuscript aims to summarize real-world evidence regarding trastuzumab deruxtecan (T-DXd) in HER2-positive and HER2-low metastatic breast cancer.Although the manuscript provides some interesting information, the quality of the manuscript is not sufficient for publication.

1- The article primarily restates findings already extensively covered in published reviews and consensus articles. There is no original synthesis, critical appraisal, or methodological innovation.

2- The majority of the text contains data tables from key DESTINY studies that have already been extensively reviewed in the high-impact literature. The "real-world" component is superficial and largely descriptive, offering no new insights or clinical guidance.

3- The literature search is poorly defined and limited to PubMed, raising concerns about completeness and bias. No PRISMA flowchart, search strategy, or inclusion/exclusion criteria are presented.

4- The majority of the article reiterates DESTINY-Breast results that are not relevant to the "real-world." This weakens the stated focus and diminishes the originality of the article.

5- Statements such as "Our analysis confirms the drug's effectiveness" are oversimplistic and not supported by a solid synthesis.

Comments on the Quality of English Language

 The English could be improved to more clearly express the research

Author Response

Comment: This manuscript aims to summarize real-world evidence regarding trastuzumab deruxtecan (T-DXd) in HER2-positive and HER2-low metastatic breast cancer. Although the manuscript provides some interesting information, the quality of the manuscript is not sufficient for publication.

Answer: we thank the reviewer for his/her comment. We tried to address and satisfy all the comments.

Comment 1: The article primarily restates findings already extensively covered in published reviews and consensus articles. There is no original synthesis, critical appraisal, or methodological innovation.

Answer: We appreciate the reviewer’s observation. Our intention with this article was to consolidate existing evidence in a concise and accessible manner for a specific audience, particularly clinicians in their daily practice, who may benefit from a focused synthesis of key findings. In particular, this is the first work in literature in which an analysis of all real-world studies with T-Dxd in MBC has been performed. That said, we recognize the value of original synthesis and critical appraisal. In response to the reviewer’s comment, we have revised the manuscript to include a more explicit critical discussion of the strengths, limitations, and gaps in the current literature. Furthermore, we have added critical comment at the end of every paragraph. We thank the reviewer for this helpful feedback, which has strengthened the focus and clarity of our work.

Comment 2: The majority of the text contains data tables from key DESTINY studies that have already been extensively reviewed in the high-impact literature. The "real-world" component is superficial and largely descriptive, offering no new insights or clinical guidance.

Answer: Thank you very much for your thoughtful feedback. We appreciate your perspective regarding the coverage of the key DESTINY studies and the real-world component of our manuscript. Our intention with the inclusion of the real-world data was to complement the existing evidence by providing a practical context and highlighting the applicability of these findings in everyday clinical settings. We aimed to offer a broader understanding, although we acknowledge that the descriptive nature may not provide novel mechanistic insights. However, we have deepened the analysis of the real-world data to enhance its clinical relevance and provide more actionable guidance. In particular, every paragraph now ends with a critical comment on the real-world data reported.

Comment 3: The literature search is poorly defined and limited to PubMed, raising concerns about completeness and bias. No PRISMA flowchart, search strategy, or inclusion/exclusion criteria are presented.

Answer: Thank you for your valuable feedback regarding the literature search methodology. As this is a narrative review, our primary aim was to provide a comprehensive overview based on key and relevant studies rather than a systematic review, which typically requires a detailed search strategy and formal reporting such as a PRISMA flowchart. However, we acknowledge the importance of transparency and have now expanded the description in the manuscript of our search process, including inclusion/exclusion criteria used to select relevant articles. We appreciate your suggestion and have revised the manuscript accordingly to clarify these points, enhancing the rigor and clarity of our review.

Comment 4: The majority of the article reiterates DESTINY-Breast results that are not relevant to the "real-world." This weakens the stated focus and diminishes the originality of the article.

Answer: Thank you for your valuable feedback. We appreciate your observation regarding the emphasis on the DESTINY-Breast results. Our intention was to provide a comprehensive background to contextualize our findings; however, we understand that the relevance to real-world application is paramount. We will revise the manuscript to better highlight the novel aspects of our study and focus more clearly on real-world implications, ensuring that the originality and practical relevance are more prominent.

Comment 5: Statements such as "Our analysis confirms the drug's effectiveness" are oversimplistic and not supported by a solid synthesis.

Answer: Thank you for your insightful comment. We acknowledge that the statement "Our analysis confirms the drug's effectiveness" may come across as overly simplistic. To address this, we have revised the manuscript to provide a more nuanced interpretation supported by a thorough synthesis of the data. We aim to present a balanced discussion that reflects the complexity of the findings and acknowledges any limitations.

Reviewer 2 Report

Comments and Suggestions for Authors
  • I feel the manuscript doesn’t focus enough on real-world data. The title makes it sound like that’s the main point, but when I read the Abstract and Introduction, most of the discussion is about clinical trials. As a result, it appears more like a summary of RCTs than a comprehensive evaluation of real-world evidence.

  • The methodology section is very brief. Just saying a “PubMed search” was done doesn’t tell the reader much. I’d like to see the actual search terms, the time period considered, and the inclusion/exclusion criteria. Even for a narrative review, this kind of detail really helps with transparency and makes it easier for others to follow your approach.

  • I noticed a few typos here and there—things like “posivite” instead of “positive,” “desease” instead of “disease,” and “intollerant” instead of “intolerant.” They’re small, but cleaning them up will make the manuscript read more professionally.

  • Some wording comes across as a bit too strong. For example, the phrase “without detrimental impact on quality of life” sounds very absolute. Real-world evidence rarely allows that kind of certainty, so a slightly softer phrasing would make the statement more accurate and credible.

  • The Discussion section leans heavily on mechanisms of action and clinical trials. Real-world findings are mentioned, but mostly in passing and descriptively. It would really help to include a more critical evaluation of these data, discussing both strengths and limitations.

  • It’s not clear how the real-world studies were selected. The text mentions nine studies, but there’s no explanation of why these were chosen or what their limitations are. Adding this information would help readers trust the conclusions more.

  • Table 3 focuses mainly on clinical trials, and the real-world findings in the text aren’t fully integrated with it. That makes the flow feel a little disjointed. Connecting these pieces more clearly would improve readability and cohesion.

  • Patient-reported outcomes and quality-of-life data are missing or not discussed in detail. Including these perspectives would give a fuller picture of how T-DXd actually affects patients in real-world settings.

  • Some sections are overly descriptive. They report outcomes like ORR, PFS, and drug mechanisms, but there’s little discussion of study limitations. Adding some critical reflection here would make the review more insightful and informative.

  • Finally, the manuscript doesn’t fully acknowledge the inherent limitations of real-world studies—things like small and heterogeneous populations, retrospective design, incomplete data, and lack of centralized tumor assessment. Explicitly addressing these would make the discussion more balanced and credible

Author Response

Comment: I feel the manuscript doesn’t focus enough on real-world data. The title makes it sound like that’s the main point, but when I read the Abstract and Introduction, most of the discussion is about clinical trials. As a result, it appears more like a summary of RCTs than a comprehensive evaluation of real-world evidence.

Answer: Thank you for your thoughtful feedback. We understand your concern regarding the balance between clinical trial data and real-world evidence in the manuscript. While clinical trials provide essential context, our primary aim is indeed to focus on real-world data. To address this, we have revised the Abstract and Introduction to better reflect the manuscript’s emphasis on real-world evidence and ensure that the discussion throughout the paper aligns more closely with this focus. We appreciate your input, which will help improve the clarity and focus of our work.

Comment: The methodology section is very brief. Just saying a “PubMed search” was done doesn’t tell the reader much. I’d like to see the actual search terms, the time period considered, and the inclusion/exclusion criteria. Even for a narrative review, this kind of detail really helps with transparency and makes it easier for others to follow your approach.

Answer: Thank you for your valuable feedback regarding the literature search methodology. We have now expanded the description in the manuscript of our search process, including inclusion/exclusion criteria used to select relevant articles. We appreciate your suggestion and have revised the manuscript accordingly to clarify these points, enhancing the rigor and clarity of our review.

Comment: I noticed a few typos here and there—things like “posivite” instead of “positive,” “desease” instead of “disease,” and “intollerant” instead of “intolerant.” They’re small, but cleaning them up will make the manuscript read more professionally.

Answer: Thank you very much for pointing out these typographical errors. We apologize for the oversights and will carefully review the manuscript to correct all such mistakes to ensure a more polished and professional presentation.

Comment: Some wording comes across as a bit too strong. For example, the phrase “without detrimental impact on quality of life” sounds very absolute. Real-world evidence rarely allows that kind of certainty, so a slightly softer phrasing would make the statement more accurate and credible.

Answer: We thank the reviewer for this comment. Considering the lack of data about quality of life in a real-life setting, we decided to remove the sentence that can lead to misunderstandings. 

Comment: The Discussion section leans heavily on mechanisms of action and clinical trials. Real-world findings are mentioned, but mostly in passing and descriptively. It would really help to include a more critical evaluation of these data, discussing both strengths and limitations.

Answer: Thank you for your insightful comment. We acknowledge that the Discussion section currently emphasizes mechanisms of action and clinical trial data. We agree that a more critical and detailed evaluation of real-world evidence, including its strengths and limitations, would enhance the manuscript. Accordingly, we have expanded the Discussion to provide a thorough analysis of real-world studies on T-Dxd in metastatic breast cancer, highlighting how these findings complement clinical trial results, while also addressing inherent challenges such as patient heterogeneity, variability in treatment settings, and data collection methods. We have also implemented the critical comments at the end of each paragraph.

Comment: It’s not clear how the real-world studies were selected. The text mentions nine studies, but there’s no explanation of why these were chosen or what their limitations are. Adding this information would help readers trust the conclusions more.

Answer: Thank you for the comment. As already said, we have now expanded the description in the manuscript of our search process, including inclusion/exclusion criteria used to select relevant articles.

Comment: Table 3 focuses mainly on clinical trials, and the real-world findings in the text aren’t fully integrated with it. That makes the flow feel a little disjointed. Connecting these pieces more clearly would improve readability and cohesion.

Answer: Thank you for your valuable feedback. We recognize that Table 3 predominantly highlights clinical trial data and it seems that there are few integrations with real-world findings. However, we decided to include also table 3 in our work to emphasize the important role of T-DXd in the treatment of MBC and to underline the evolving landscape and the necessity of integrate real-world data with data from clinical trial to improve daily patient’s management.

Comment: Patient-reported outcomes and quality-of-life data are missing or not discussed in detail. Including these perspectives would give a fuller picture of how T-DXd actually affects patients in real-world settings.

Answer: Thank you for your valuable comment. As noted in the manuscript, patient-reported outcomes and quality-of-life data are currently limited or lacking in the available real-world studies on T-DXd. We acknowledge the importance of these perspectives and have highlighted this gap to emphasize the need for further research to better understand the impact of T-DXd on patients’ quality of life in real-world settings.

Comment: Some sections are overly descriptive. They report outcomes like ORR, PFS, and drug mechanisms, but there’s little discussion of study limitations. Adding some critical reflection here would make the review more insightful and informative.

Answer: Thank you for your thoughtful comment. We appreciate the suggestion to include a more critical reflection on study limitations. In the Discussion section, we have addressed the limitations of the real-world studies reviewed, highlighting aspects such as small sample sizes, potential biases, and variability in data collection. We will further strengthen this section to provide a deeper and more balanced analysis, enhancing the overall insightfulness of the review.

Comment: Finally, the manuscript doesn’t fully acknowledge the inherent limitations of real-world studies—things like small and heterogeneous populations, retrospective design, incomplete data, and lack of centralized tumor assessment. Explicitly addressing these would make the discussion more balanced and credible.

Answer: Thank you for your insightful comment. We agree that a thorough acknowledgment of the inherent limitations of real-world studies is crucial to provide a balanced and credible discussion. In our revision, we have explicitly addressed key challenges such as the typically small and heterogeneous patient populations, the retrospective nature of many studies, incomplete or missing data, and the lack of centralized tumor assessments that may affect consistency and reliability of outcomes. Highlighting these limitations will help contextualize the findings, clarify the scope of real-world evidence, and underscore the need for cautious interpretation when comparing with clinical trial data. We appreciate this suggestion and will incorporate it to strengthen the overall rigor and transparency of our manuscript.

Reviewer 3 Report

Comments and Suggestions for Authors

The review describes the response to treatment with trastuzumab deruxtecan  and its side effects in advanced HER2- positive/HER2 – low breast cancer. Since trastuzumab deruxtecan is a new generation of breast cancer therapy, it is important to evaluate the outcome of the treatment. The authors also focused on breast cancer patients with metastases. It is very important group of patients. There is good screening for breast cancer and most of the cases are diagnosed in early stage where survival rate is very high. So, the analysis of survival and adverse effects after ADC therapy showing that the results of the treatment are better than with another regimen which were used earlier, is important. The following paragraphs: heavily pretreated patients, poor ECOG – PS patients, HER2-low versus HER2 – positive, brain metastasis, elderly, interstitial lung disease, describe the different groups of patients and compare the outcome of the treatment with ADC and it should be very important for chemotherapists treating breast cancer patients.

The manuscript is very interesting since it describes the outcome of modern ADC therapy for advanced breast cancer patients. Substantial part of the manuscript is devoted to adverse effects of therapy with the information that ADC therapy causes manageable side effects.

Author Response

The review describes the response to treatment with trastuzumab deruxtecan  and its side effects in advanced HER2- positive/HER2 – low breast cancer. Since trastuzumab deruxtecan is a new generation of breast cancer therapy, it is important to evaluate the outcome of the treatment. The authors also focused on breast cancer patients with metastases. It is very important group of patients. There is good screening for breast cancer and most of the cases are diagnosed in early stage where survival rate is very high. So, the analysis of survival and adverse effects after ADC therapy showing that the results of the treatment are better than with another regimen which were used earlier, is important. The following paragraphs: heavily pretreated patients, poor ECOG – PS patients, HER2-low versus HER2 – positive, brain metastasis, elderly, interstitial lung disease, describe the different groups of patients and compare the outcome of the treatment with ADC and it should be very important for chemotherapists treating breast cancer patients.

Comment: The manuscript is very interesting since it describes the outcome of modern ADC therapy for advanced breast cancer patients. Substantial part of the manuscript is devoted to adverse effects of therapy with the information that ADC therapy causes manageable side effects.

Answer: we thank the reviewer for his/her positive comment.

Round 2

Reviewer 1 Report

Comments and Suggestions for Authors

I am satisfied that the authors have addressed all of my previous concerns about the article. It is now much improved and I feel that it is now suitable for publication. 

Reviewer 2 Report

Comments and Suggestions for Authors

Thank you for your responses and the effort you put into improving the manuscript “Trastuzumab deruxtecan for the treatment of metastatic breast cancer patients: data from real-world studies.” I believe this version is suitable for publication, and I accept the manuscript in its current form.